# Development of Supervised Speaker Diarization System Based on the PyAnnote Audio Processing Library

**DOI:** 10.3390/s23042082

**Published:** 2023-02-13

**Authors:** Volodymyr Khoma, Yuriy Khoma, Vitalii Brydinskyi, Alexander Konovalov

**Affiliations:** 1Faculty of Electrical Engineering, Automatic Control and Informatics, Opole University of Technology, 45-758 Opole, Poland; 2Institute of Computer Technologies, Automation and Metrology, Lviv Polytechnic National University, 79013 Lviv, Ukraine; 3Vidby AG, Suurstoffi 8, 6343 Rotkreuz, Switzerland

**Keywords:** diarization system, PyAnnote library, identification of speakers’ utterances, segmental classification, cluster classification

## Abstract

Diarization is an important task when work with audiodata is executed, as it provides a solution to the problem related to the need of dividing one analyzed call recording into several speech recordings, each of which belongs to one speaker. Diarization systems segment audio recordings by defining the time boundaries of utterances, and typically use unsupervised methods to group utterances belonging to individual speakers, but do not answer the question “who is speaking?” On the other hand, there are biometric systems that identify individuals on the basis of their voices, but such systems are designed with the prerequisite that only one speaker is present in the analyzed audio recording. However, some applications involve the need to identify multiple speakers that interact freely in an audio recording. This paper proposes two architectures of speaker identification systems based on a combination of diarization and identification methods, which operate on the basis of segment-level or group-level classification. The open-source PyAnnote framework was used to develop the system. The performance of the speaker identification system was verified through the application of the AMI Corpus open-source audio database, which contains 100 h of annotated and transcribed audio and video data. The research method consisted of four experiments to select the best-performing supervised diarization algorithms on the basis of PyAnnote. The first experiment was designed to investigate how the selection of the distance function between vector embedding affects the reliability of identification of a speaker’s utterance in a segment-level classification architecture. The second experiment examines the architecture of cluster-centroid (group-level) classification, i.e., the selection of the best clustering and classification methods. The third experiment investigates the impact of different segmentation algorithms on the accuracy of identifying speaker utterances, and the fourth examines embedding window sizes. Experimental results demonstrated that the group-level approach offered better identification results were compared to the segment-level approach, and the latter had the advantage of real-time processing.

## 1. Introduction

Diarization is an important step in the process of speech recognition, as it partitions an input audio recording into several speech recordings, each of which belongs to a single speaker. Traditionally, diarization combines the segmentation of an audio recording into individual utterances and the clustering of the resulting segments. Segmentation identifies points where speakers change in an audio recording, and the clustering groups speech segments on the basis of the main characteristics of the speaker [1].

Although the development of diarization methods started more than a decade ago, intensive research continues to improve the accuracy and computational efficiency of diarization algorithms. Diarization systems mainly utilize unsupervised machine learning algorithms when utterances are shared between speakers, but it is not known which diarization label applies to a particular speaker. This approach is known as unsupervised diarization. However, in some applications, it is necessary to identify multiple speakers that interact freely in an audio recording. This task can be completed using a supervised diarization approach, which combines diarization and identification methods.

The main distinction between unsupervised and supervised diarization involves the different ways of segment indexation. The former relies on grouping of similar segments into separate categories (clustering), while the latter requires matching a segment with a certain speaker on the basis of voice samples (classification). Thus, the last transformation step for supervised diarization is performed with the help of an additional output module—the speaker‘s classifier. However, the identification of the speaker using supervised diarization has a significant peculiarity, as it is necessary, unlike in the common biometric voice recognition systems, to analyze the utterances of more than one subjects. This is a challenge, especially under low-data conditions, as new speakers are represented only by one short audio recording lasting dozens of seconds at most.

The application of unsupervised diarization allows for improving the readability of automatic speech-to-text transcription due to the structure of the audio recording upon the queue of speakers [2]. Another area where unsupervised diarization can be applied is in long audio recordings with several speakers, where diarization is used to divide a long audio recording into shorter ones before actual speech recognition takes place [3]. It is also used in translation systems, where an audio recording is split into speech segments with the help of diarization, and the system performs translation for each segment, thus increasing the speed and accuracy of such systems [4,5].

Companies and individuals are increasingly interested not only in the audio transcription of conversations coming from online meetings and video conferences, but also in the automatic identification of speakers. These objectives can be achieved using supervised diarization systems, which allow for the identification of speech segments belonging to a particular speaker. Thus, it becomes possible to perform the voice search/indexation of content within recorded conversations [6].

A prevalent supervised diarization use case is the annotation of important phone calls. For such a purpose, a conversation recording of a caller and an operator is partitioned into two speech clips using diarization. Afterwards, those clips are sent to a speech recognition system that provides corresponding speech-to-text transcription. Another exemplary use of supervised diarization is in healthcare and medical services, where the identity of a doctor is known beforehand [7]. This allows for focusing on a doctor’s speech processing, and thus end up with more reliable and accurate results for different use cases, such as the detection and analysis of diagnosis, prescriptions, and procedures.

In the developing modern digital world, application scenarios of diarization systems are constantly expanding, which often appear as components of more complex information systems in human–machine communication. Therefore, research aimed at the development and implementation of a flexible and open-source system capable of solving the main problems of diarization, both in a supervised and unsupervised manner, is relevant. This article presents the architecture of a diarization system using the PyAnnote framework combined with external components that provide expanded functionality.

## 2. Materials and Methods

### 2.1. Literature Review

A review of literature sources gives grounds to state that diarization algorithms are traditionally based on an unsupervised approach [8,9,10]. Stagewise speaker diarization architectures containing a sequence of modules such as speech detection, speech segmentation, embedding extraction, clustering, and labeling clusters have been studied for a long time [2,10,11]. Moreover, end-to-end diarization technology is being intensively developed, which allows for increasing the accuracy of diarization by optimizing not individual modules, but the system as a whole, in particular by using selected supervised speaker clustering methods [12,13,14].

Although speaker diarization is usually an audio-related task, a separate field of research, multimodal speaker diarization, has arisen. These systems use some information in addition to the audio to improve diarization quality [15]. These may be, for example, the speaker’s behavioral characteristics extracted from a synchronized video [16] or linguistic content conveyed by speech cues [17].

A number of studies were devoted to the design of systems utilized for the identification of statements in audio recordings belonging to multiple speakers [18,19,20]. Similar systems combine features of diarization and identification systems. Traditional voice-based biometric systems typically identify a person by analyzing an audio signal/recording where only one speaker is present. Therefore, other approaches need to be developed to identify speakers and their statements if more than one speaker are arbitrarily present in the audio signal.

Most diarization systems work offline, that is, the diarization result is obtained after analyzing the entire audio recording. Online audio diarization is able to output the diarization result of each audio segment immediately after its analysis. Maintaining online diarization accuracy is a challenging task due to the unavailability of data from future audio segments. The creation of online diarization systems opens up new possibilities and extends the range of applications of modern speech processing technologies [11,21,22].

Diarization systems are used to perform a number of speech processing tasks in various scenarios. For their implementation, proprietary solutions can be used, primarily from such technological IT giants as IBM (IBM Watson) or Google (Google Cloud Speech) [23]. Nevertheless, it is also possible to use open-source libraries, for example, pyAudioAnalysis [24], SpeechBrain [25], or PyAnnote [10].

Commercial products are first designed as solutions for speech recognition, whereas diarization is a secondary feature. Advantages of such solutions are the simplicity of usage, the speed of development, and system adjustment rather than good accuracy. However, there are drawbacks as well: no possibility to adjust the system parameters or its separate components, the price, and a dependency on the infrastructure of the service provider.

Taking into account the variety of use cases and trends mentioned in the above design and implementation of a general-purpose system capable of solving both supervised and unsupervised major diarization tasks is a relevant research topic from both theoretical and practical perspectives. It also makes sense to design such a system so that it could process audio data both in online and offline modes. It is reasonable to carry out development on the basis of an open-source framework with a stagewise architecture, as it opens the possibility to reuse existing components and adjust the system’s parameters. Another important point is that the selected framework should demonstrate results comparable to those of commercial products in terms of accuracy. After deeper analysis, the following frameworks were identified as satisfying the above-mentioned demands: PyAnnote, SpeechBrain, and pyAudioAnalysis.

PyAudioAnalysis is an open-source library designed for sound processing, namely, for feature extraction from audio recordings with their further classification and segmentation. Thus, such a library may be used for the purposes of supervised diarization. This library has a number of designed tools. First, data analysis and feature extraction from the audio recordings. It also has tools for audio segmentation and classification, but its accuracy is significantly lower than that of the commercial solutions and other open-source solutions [10].

SpeechBrain—is a set of open-source software tools designed for creating an audio analysis system, namely for such tasks as automatic speech recognition, speaker identification, language identification, etc. Currently, this library does not contain any tool for audio segmentation which complicates its application for the diarization tasks [25].

PyAnnote is a directly developed open-source library for diarization tasks [3,10,26]. It consists of trained neural network modules on which an unsupervised diarization system can be implemented. The above-mentioned modules may be additionally readjusted, substituted with other modules, or added with new ones. Such an open-module structure provides high flexibility and possibilities for the improvement of the system’s parameters. Moreover, this library contains tools for audio processing and manipulation [10].

Thus, the authors selected PyAnnote for further research, as it provides the highest accuracy (compared with existing commercial solutions) among the available options and is implemented using stagewise modular approach which enables direct hyperparameters’ tuning. Thus, the core requirements formulated above are fulfilled, and PyAnnote can be considered a suitable solution for supervised and unsupervised diarization.

### 2.2. Research Aim

The aim of this work was to design a general-purpose diarization system (capable of operating in supervised and unsupervised mode) using the PyAnnote framework. This requires adding extra modules for speaker identification along an adjustment of the system’s parameters on the stagewise level in order to extend its functionality and improve the diarization results.

### 2.3. PyAnnote—Speaker Diarization Library

The current research basically uses the unsupervised diarization system (using clustering) on the basis of PyAnnote (version 1.1) presented in Figure 1 [10].

The following modules were included in the system composition:Segmentation—module that uses the voice activity detection, speaker change detection, and overlapped speech detection modules to create timestamps that represent segments of speech within an audio recording.(a)Voice activity detection—the module’s function is to detect the time intervals in the audio recording where a human voice is present.(b)Speaker change detection—the module’s function is to detect the moments of time in the audio recording where the speech of one speaker ends and the speech of another one starts.(c)Overlapped speech detection—the module’s function is to detect time intervals in the audio recording where two or more speakers are talking simultaneously. Such segments are removed from analysis, as, in such cases, additional transformations related to the separation of signals overlapping in time are performed to detect a particular speaker.Speaker embedding—a module to create a vector with a certain given dimension, of which the numerical values represent speaker-specific features derived on the basis of physiological voice parameters. Thus, the distance between vectors that correspond to phrases belonging to one person is smaller than the distance to the vectors created from the phrases spoken by the other speakers.Clustering—the module responsible for grouping segments corresponding to the speakers by using their embeddings.

### 2.4. Suggested Approach

At the output of an unsupervised diarization system (Figure 1), the input audio recording is divided into segments and grouped into clusters, but it is unknown which cluster belongs to which speaker. We researched the idea of adapting the PyAnnote for the purposes of the supervised diarization tasks by adding a classifier to the system. This modification allows for the identification of speakers or, in other words, to assign a particular group of audio segments to the particular speaker. In this regard, we propose two architectures of a supervised diarization system (SDS):Architecture A (Figure 2, SDS A) was designed by substituting the clustering module with the classification module in the basic architecture of an unsupervised diarization system (Figure 1).Architecture B (Figure 2, SDS B) was designed by adding the classification module to the basic architecture of an unsupervised diarization system (Figure 1), i.e., by combining the implementation of the two modules (clusterization and classification).

In order to select a particular implementation of the classification algorithm, in a real-world application, supervised diarization systems are constantly tuned in order to support new speakers. Normally, the amount of available data for such tuning is very small (few short phrases at best), which is not enough to train common classification models such as XGBoost, neural networks, and SVM. In our research, 20 s of reference speech was used to add a new user to the diarization system. Due to this constraint, we built custom classifier models using a distance function followed by a threshold. This was applied to both architectures presented in Figure 2.

Type A architecture is based on straightforward segmentwise classification. Each segment is matched to a particular speaker on the basis of the distance between the reference embeddings of the speaker and the embeddings of the certain audio segment. Depending on the previously adopted threshold, it is defined whether the segment belongs to the speaker or not. This method is suitable for real-time or online diarization, which means that it can also be used with audio streams and not only audio recordings. Grid search was used for threshold selection. The selected thresholds for the distance functions are the following: Euclidean distance threshold - 7.0; Cosine distance threshold - 0.8; Manhattan distance threshold - 5.0; Pearson’s distance threshold - 0.5; Spearman’s distance threshold - 0.5. Type B architecture is based on the idea of the distance to the group (cluster centroid). In this case, before choosing which segments refer to a particular speaker, all segments are grouped, that is, unsupervised diarization is performed in essence. After audio-segment embeddings are grouped, the centroid for each group is defined (the mean value of all the embeddings in the group); afterwards, the distance from the target speaker embedding to each of the centroids is calculated, the closest centroid is selected, and the audio segments of its corresponding group are marked as the segments belonging to the target speaker. This method is suitable for offline diarization only (no real-time capabilities) because it performs the analysis of the entire speech recording as a whole.

## 3. Results

In order to implement the approach described in the previous section, a set of optimization experiments was performed. We decided to study the impact of separate modules and their parameters on the final results of supervised diarization. Experiments were performed in the following order:Classifier design using distance functions (Architecture A).Classifier design using clustering algorithm and distance functions (Architecture B).Segmentation optimization to maximize Fscore and minimize DER.Embedding window optimization for the maximization of the F score and the minimization of DER.

The reason for this was to design new components that eere related to supervised diarization, first by using default configuration from unsupervised pipeline for segmentation and embedding modules. Afterwards, an additional optimization for the segmentation and embedding part is run in order to boost the accuracy of supervised pipeline.

After the implementation and optimization of the supervised pipeline, its results were compared with those of an original unsupervised pipeline. For this purpose, the following default hyperparameters provided by the PyAnnote developers were used:Voice activity detection: sampling_rate=16 kHz, input_size=32,000 datapoints, thresholds for offset=0.48 and onset=0.48.Speaker change detection: sampling_rate=16 kHz, input_size=32,000 datapoints, threshold = 0.146.Embeddings: DNN x-vectors, vector_length=512 units, input chunck_size=2.0 s.Clustering: probabilistic linear discriminant analysis (PLDA).

All the experiments were conducted on the same dataset—AMI Corpus, which is an open-source dataset containing audio and video recordings of meetings and gatherings where speech is annotated and each annotation segment is assigned with the identification number of the speaker; thus, these data are suitable for both supervised and unsupervised diarization [27]. Audio data are stored in the WAV format, annotation in the XML format, which is transformed into Rich Transcription Time Marked (RTTM) format for work with the PyAnnote library. RTTM is the text format containing information about segments and speaker identification for each of the audio files. The size of the dataset was 100 h. The average duration of one audio file was 2104.48 s. The average duration of one annotated segment was 4.1 s. The number of unique speakers was 189 (65.1% of them were males). The average duration of audio per speaker was 546.96 s.

Experiments for the supervised diarization were also conducted for each speaker in each audio recording. For example, an audio recording with 4 speakers was analyzed in turns, and accuracy was evaluated separately for each participant. When the number of speakers is not given, the system must determine it. Ground-truth data are given, and performance metrics are calculated on the basis of unseen data.

The following metrics are widely used to evaluate diarization systems [26]: detection F score, segmentation F score, diarization error rate, and identification F score.

Detection F score—the accuracy score for voice detection in the audio recording.Segmentation F score—the accuracy score for the detection of segment boundaries and overlaps in the audio recording.Diarization error rate—the error of detecting a segment’s boundaries and overlaps in an audio recording considering the true or false assignment of the speaker identifier to the audio recording segment. This is the error main for diarization and is the generally accepted metric in commercial systems.Identification F sscore—the accuracy score for the identification of speakers on each segment in an audio recording, which is calculated considering the duration of each segment.

The identification F score (or F1 measure) was selected as the main metric, as it is the one that correctly reflects the accuracy of assigning the speaker identifier to each of the audio recording segments, unlike the diarization error, which uses automatic segment correlation that is not connected to any of the speakers. The diarization error rate was selected as a secondary metric to enable comparison between supervised and unsupervised pipelines.

### 3.1. Experiment 1: Classifier Design Using Distance Functions

The purpose of this experiment was to study the impact of different distance functions in the context of the supervised diarization accuracy (DER and F-Score). The principle of this distance-based classification is quite simple. We had a built-in reference speaker that played audio received from the reference speaker audio. Further, for each segment in the audio recording, we calculated the distance between the reference embedding and the analyzed segment embedding. If the distance was lower than the defined threshold, the segment belonged to the reference speaker. The following distance functions were studied: Euclidean, Manhattan, Pearson’s, Spearman’s, and cosine [28,29] because they are widely used in machine-learning and various identification tasks, and they are available in the scikit-learn library. The results of the experiment are presented in Table 1.

The experiment results show that cosine distance had the best result, just slightly outperforming Pearson’s and Spearman’s distances.

### 3.2. Experiment 2: Classifier Design Using Clustering Algorithm and Distance Functions

The goal of this experiment was to determine the impact of various clustering techniques on supervised diarization performance. The experiment flow was the following:Clustering all the audio segments embeddings.Calculating each cluster centroid.Calculating the distance from the reference speaker embedding to each of these centroids, and the closest distance determines which cluster contains segments with the reference speaker.

The following clustering algorithms were chosen: PLDA (default for an unsupervised pipeline), K-means, hierarchical, and spectral because they are the most popular clustering algorithms, and their implementations are available in the scikit-learn library. Taking into account the results of the previous experiments, the cosine function was used for centroid classification. Experimental results are presented in Table 2.

This table shows that K-means clustering dramatically outperformed both the hierarchical and spectral algorithms. A combination of K-means clustering with cosine distance also demonstrated much better results than those of segment-wise classification presented in Experiment 1. Thus, Architecture B was better in terms of diarization accuracy than Architecture A. However, as mentioned earlier, it is not suitable for real-time applications and can be utilized for offline processing only.

### 3.3. Experiment 3: Segmentation Optimization

The goal of this experiment was to determine how different segmentation algorithms affect the performance of a supervised diarization pipeline. Taking into consideration the available building blocks, the following modifications of the default PyAnnote segmentation methods are proposed:Standard—the default PyAnnote segmentation that relies on a combination of the voice activity detection (VAD) and speaker change detection (SCD) modules. This implementation is two-stage. First, VAD separates speech and nonspeech segments. Afterwards, speech segments are additionally processed by SCD, which splits them into single-speaker segments.Overlap drop-out—in addition to VAD and SCD, this approach also uses overlap detection (also provided by PyAnnote) as the last transformation in the segmentation pipeline. If voices are quite similar, there is a greater chance that the speaker change detection would not be able to correctly identify the boundaries. However, after applying overlap detection, such sections can be identified as overlapped speech and can be effectively removed. This results in increased chances of finding speaker change boundaries and thus improving diarization performance.

Optimization was performed separately for the two architectures from Figure 2. Experimental results are presented in the Table 3.

The experimental results show that the standard segmentation method just slightly outperformed overlap drop out while being less computationally expensive. So, we continued with the standard PyAnnote segmentation for both SDS architectures.

### 3.4. Experiment 4: Embedding Optimization

The goal of this experiment was to determine the optimal size of the embedding window. Embeddings were calculated for each individual segment using a rolling window (with no overlap) followed by averaging. For the embedding extraction, the default PyAnnote embedding model was used. It has a tuneable parameter called chunk_size that controls the embedding window size. Optimization was performed separately for the two architectures from Figure 2. Experimental results are presented in the Table 4.

The performance of supervised diarization was clearly correlated with the embedding window size: the lower the window was, the better the performance. With a chunk size of 1.0 s, the supervised diarization performance increased by almost 4.5% for the supervised architecture of Architecture A, compared to the default chunk size of 2.0 s. The performance increase of supervised Architecture B was negligible compared to the default chunk size, only 0.1%. The larger chunks only worsened the supervised diarization performance.

## 4. Discussion

To compare the pipelines, the diarization error rate (DER) metric was chosen, which is widely used in speaker diarization tasks. It is measured as the fraction of time that is not attributed correctly to a speaker or nonspeech. This metric does not, however, require to correctly identify the speakers by definite identification number or name, which renders it suitable to compare unsupervised and supervised speaker diarization systems. For supervised speaker diarization, the F score metric is calculated, as it better reflects the errors of labeling speakers with a wrong ID. There is also a strong negative correlation between F score and DER (the lower the DER is, the higher the F score).

The following set of parameters were selected for the SDS of the Type A architecture: cosine distance followed by thresholding as a classification model, embedding window size of 1.0 s, and segmentation based on VAD, SCD, and overlap detection. The length of the reference speaker audio per class is 20 s.

For the SDS of the Type B architecture, the following configuration was proposed: cosine distance followed by thresholding as a classification model, K-means as clustering algorithm followed by centroid calculation, embedding window size of 1.0 s, and segmentation based on VAD, SCD, and overlap detection. The length of the reference speaker audio per class is 20 s.

The baseline (unsupervised) PyAnnote pipeline from Figure 1 yielded a DER of 24.97%.

The segmentwise supervised diarization pipeline (type A) yielded a DER of 20.14%, which was an improvement over the baseline by 4.84%, and an F score of 85.89%.

The group-based supervised diarization pipeline (Type B) yielded a DER of 15.52%, which was an improvement over the baseline by 9.45%, and an F score of 90.79%.

The main sources of performance improvement included the clustering algorithm and distance function. The best clustering algorithm and distance function for the task were the K-means algorithm and cosine distance, respectively. Another major source of improvement was the transition from the unsupervised to supervised method of diarization. Since the model knew the speaker embeddings beforehand, it was easier and more accurate to assign the segments to the correct speaker.

Additionally, some steps can be taken to further improve diarization performance: filtering acoustic disturbances, i.e., other nonspeech and environmental sounds, perhaps by using speech separation techniques or other signal processing methods; addressing diction fluctuations or different speaker expressions where one speaker with different expressions could be identified as a different speaker; filtering segment edge effects where speech begins or ends (or briefly overlaps with another speaker), which may affect the speaker embedding, which in turn affects the classification performance; preventig a speech segment from being too short or too long, which also affects the speaker embedding of a given segment.

## 5. Conclusions

PyAnnote is one of the most common and functional solutions in the field of audio signal processing, particularly involving diarization. Its main advantages over well-known alternatives are implementation in the popular Python language, open-source code under an MIT license, modular architecture with the possibility of adding new modules to the system, and adjusting the parameters of existing components. In addition, PyAnnote is almost as accurate as existing commercial solutions such as IBM Watson or Google Speech.

The existing implementations of the PyAnnote library are designed primarily for tasks of unsupervised diarization. At the same time, there is the opportunity to expand the functionality of the system and adapt PyAnnote to tasks of supervised diarization, which offers new potential for speaker identification and searching an audio database for recorded parts belonging to a specific speaker. A new architecture design and the parameter optimization of audio processing pipeline (segmentation, embeddings extraction, clustering and classification) was the subject of research.

The approach proposed in this paper provides the possibility to select between two options for speaker identification using the PyAnnote framework. The first operates in a segmentwise fashion by substituting the clustering module from the original pipeline with a classifier. The second is based on the combination of clustering and classification. In fact, in the first case, speaker identification is performed at the level of individual audio segments, while in the second case, identification occurs on the basis of groups of segments of the same type combined into clusters.

In order to investigate the proposed approach for speaker identification and optimize the separate components of the diarization pipeline, four experiments were conducted. All research was conducted on an open dataset, AMI Corpus, which contains the marked audio and video recordings of working meetings and meetings. Metrics such as F score and diarization error rate were used for pipeline evaluation.

The first experiment investigated the accuracy of diarization under different implementation classification algorithms for segment-level architecture. In the second experiment, research was carried out to select the optimal clustering algorithm for the group-based architecture of a supervised diarization system. In the third experiment, we examined and selected the optimal segmentation algorithm for the two architectures. The fourth experiment was aimed at choosing the optimal duration of the window of vector embeddings, which is necessary for the optimal vector representation of speakers within audio recording segments.

The proposed approach enabled expanding the functionality of the speaker diarization system based on PyAnnote audio processing by adding three operational modes:Unsupervised mode: ready to be used out of the box, as no additional activities are required to add a new speaker into the system with DER of about 24.97%.Segmentwise supervised mode: opens up the possibility of real-time operation for speaker identification, and provides a DER of 20.14% and an F value of 85.89%.Group-based supervised mode: provides the highest accuracy yields, a DER of 15.52% and an F value of 90.79%.

In the future, it is appropriate to conduct additional research in order to optimize other modules of the PyAnnote (voice activity detector, speaker change detector, speech overlap detector, embeddings extraction algorithm) to further improve diarization accuracy.

## Figures and Tables

**Figure 1 sensors-23-02082-f001:**
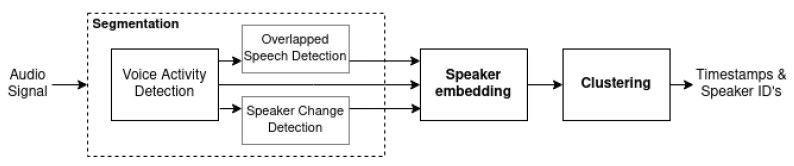
Basic structure of the diarization system with clustering based on PyAnnote.

**Figure 2 sensors-23-02082-f002:**
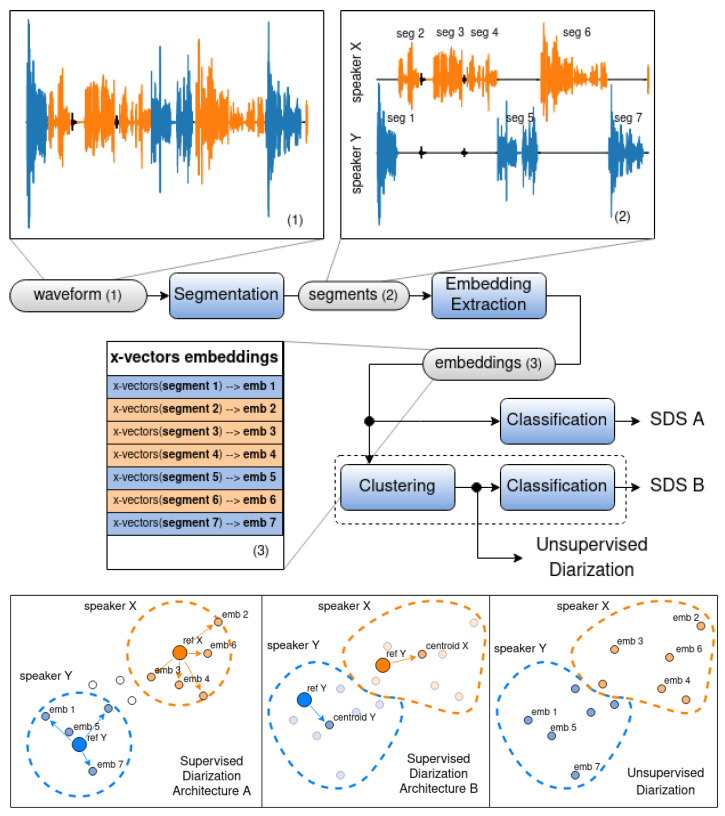
Architectures of the general-purpose diarization systems based on PyAnnote: **(Architecture A)** identification via separate segments, **(Architecture B)** identification based on the group (cluster) of segments, and unsupervised segment clustering.

**Table 1 sensors-23-02082-t001:** Experiment 1 results.

Model	Distance Function	Diarization Error Rate, %	F Score, %
Supervised A	Euclidean	30.12	75.29
**Supervised A**	**cosine**	**24.74**	**81.00**
Supervised A	Manhattan	42.77	61.80
Supervised A	Pearson’s	25.19	80.78
Supervised A	Spearman’s	25.99	80.33

**Table 2 sensors-23-02082-t002:** Experiment 2 results.

Model	Clustering Method	Diarization Error Rate, %	F Score %
Supervised B	PLDA	18.69	87.40
**Supervised B**	**K-Means**	**15.62**	**90.69**
Supervised B	Hierarchical	42.60	61.98
Supervised B	Spectral	36.20	67.94

**Table 3 sensors-23-02082-t003:** Experiment 3 results.

Model	Segmentation Method	Diarization Error Rate, %	F-Score, %
**Supervised A**	**Standard**	**24.74**	**81.00**
Supervised A	Overlap drop-out	25.49	80.75
**Supervised B**	**Standard**	**15.62**	**90.69**
Supervised B	Overlap drop-out	16.07	89.80

**Table 4 sensors-23-02082-t004:** Experiment 4 results.

Model	Embedding Chunk Size, s	Diarization Error Rate, %	F-Score, %
Supervised A	0.5	22.60	83.28
**Supervised A**	**1.0**	**20.14**	**85.89**
Supervised A	1.5	20.85	85.13
Supervised A	2.0 (default)	24.74	81.00
Supervised A	3.0	27.63	77.90
Supervised A	4.0	30.13	75.25
Supervised B	0.5	22.15	82.40
**Supervised B**	**1.0**	**15.52**	**90.79**
Supervised B	1.5	15.56	90.75
Supervised B	2.0 (default)	15.62	90.69
Supervised B	3.0	15.71	90.59
Supervised B	4.0	21.53	81.82

## Data Availability

Data are available in a publicly accessible repository that does not issue DOIs. Publicly available datasets were analyzed in this study. These data can be found here: https://groups.inf.ed.ac.uk/ami/corpus/, accessed on 24 November 2022.

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
