# Peer review of "Development of Supervised Speaker Diarization System Based on the PyAnnote Audio Processing Library"

_sensors, 2023, doi:10.3390/s23042082_

Round 1

Reviewer 1 Report

Table. 1 ~ Table. 4 in manuscript has same description. It should be modified to describe corresponding experiments.

According to reference [10] (PYANNOTE.AUDIO: NEURAL BUILDING BLOCKS FOR SPEAKER DIARIZATION) and PyAnnote github, evaluated DER for AMI dataset using default pipeline of Pyannote 1.1 is 29.7%. However, in discuss section of authors manuscript, authors described Pyannote pipeline yields a DER of 24.97%. Is it just mislead of numbers or are there any reason Pyannote performs presented DER?

( Yes, I know lower DER is better,  and presented PyAnnote's DER in the manuscript [24.97] has better performance than original Pyannotes DER [29.7],  the source of numbers should be clear.)

Overall, the manuscript is well written and the authors method(s) achieved better performing speaker diarization pipeline for PyAnnote than existing one.

Author Response

 Dear Reviewer,
Thank you for taking your time and revision our manuscript.
We have made corrections to the paper.
Best Regards,
Authors

Comments and Suggestions for Authors
1) Table. 1 ~ Table. 4 in manuscript has same description. It should be modified to describe corresponding experiments.
Answer: Updated in the manuscript
2) According to reference [10] (PYANNOTE.AUDIO: NEURAL BUILDING BLOCKS FOR SPEAKER DIARIZATION) and PyAnnote github, evaluated DER for
AMI dataset using default pipeline of Pyannote 1.1 is
29.7%. However, in discuss section of authors manuscript, authors described Pyannote pipeline yields
a DER of 24.97%. Is it just mislead of numbers or are there any reason Pyannote performs presented DER?
( Yes, I know lower DER is better, and presented PyAnnote's DER in the manuscript [24.97] has better performance than original Pyannotes DER [29.7], the source
of numbers should be clear.)
Answer:
The reason that numbers vary is that in the original paper evaluation was performed on the entire dataset as it considered only unsupervised
learning. In our case evaluation was performed on the test set only to make a fair comparison between supervised and unsupervised pipelines.
Overall, the manuscript is well written and the authors method(s) achieved better performing speaker diarization pipeline for PyAnnote than existing one.
Answer:
Many thanks to the Reviewer 1

Reviewer 2 Report

The contribution of the paper is to propose two architectures of speaker  identification systems based on a combination of diarization and identification methods, which operate based on segment-level or group-level classification. Experimental results demonstrated that the group-level approach offered better identification results compared to the segment-level approach, but the latter has the advantage of real-time processing.

In conclusion, my opinion on the paper is positive.

The contents of the paper is interesting and academically worthwhile.

The main results in the paper are correct as far as I can determine and interesting enough to be published.

Thus, I am happy to recommend that the paper is accepted for publication in Sensors.

Author Response

Dear Reviewer,
Thank you for taking your time and revision our manuscript.
Best Regards,
Authors

Reviewer 3 Report

The authors developed a supervised speaker diarization system.

However, 1. the new method has to be compared with other methods for the preparation of the introduction.

2. Table 1,  what is the F score? F1 is more commonly used.

3. The title is "Development", but the paper failed to introduce the development details: hard & soft platform, program design, UI and so on.

4. The topic is novel but the application proposed is not so novel. Please rewrite the conclusions.

For these reasons, I suggest to minor revision.

Author Response

 Dear Reviewer,
Thank you for taking your time and revision our manuscript.
We have made corrections to the paper. We hope that you will find our updated paper better.
Best Regards,
Authors

Reviewer 4 Report

The authors present a work that leverages an open-source PyAnnote to design the speaker diarization and identification of audio data. They also proposed two approaches: 1) an online method based on thresholds and 2) an offline method based on clustering. This manuscript is well written. Therefore, this reviewer provides some minor comments.

  1. Introduction: The authors mainly discuss the difference between supervised and unsupervised approaches in this research area. The introduction section fails to include the reviews of the state-of-the-art technologies and their limitations in relevant research areas while including the reviews in the following subsection. The authors might include a belief summary of the current practice and limitations to attract readers' attention. 
  2. Introduction: The authors need to claim how the authors' work can contribute to the related research field. For example, the authors need to argue how the proposed system can contribute to some applications with improved capabilities. Does this system create a new scientific capability on top of the existing software and algorithm developments?
  3. Introduction: The authors need to provide a scope of this manuscript at the end.
  4. Figure 1: What are the outcomes of software architecture? The figure has a right arrow at the end of the right side. The authors might include the outcomes of the proposed framework.
  5. 2.4. Suggested approach: The authors need to include more details in this section. The current description is too general. The authors need to introduce 1) hyperparameters that the authors aim to test, 2) what is the role and impact of each hyperparameter on the outcomes of the system, 3) need to explain how to extract embeddings from the signal and the size of the embedding vectors, and 4) need to explain the performance metrics that the authors are using.
  6. Furthermore, the authors need to provide a problem description here. Some of the underlying conditions are not clear. For example, the number of speakers is given or not, grout truth data are given, and cross-validation setups (are the performance metrics calculated based on unseen data?).
  7. Lines 227-232: The authors need to provide the definition of offset, onset, threshold, chunck_size. In the bullet points, the authors mention PLDA. However, the authors tested several clustering algorithms.
  8. Line 272: Need references for Euclidean, Manhattan, Pearson's, Spearman's, and cosine
  9. Section 3.1. Classifier design using distance functions (architecture A) is based on the thresholds, which rely on the distance functions. Therefore, the error rates and f-scores can depend on the selection of thresholds. The authors need to provide how the authors determine the thresholds, and the thresholds must be shown in Table 1. Furthermore, Architecture A is an online algorithm: therefore, the performance metrics can change over time. The authors might provide a figure that shows the convergence of the performance metrics over time.
  10. Sections 3.2. Hierarchical and Spectra algorithms might have more hyperparameters on top of the definition of distance functions. The authors need to provide which hyperparameters were used in this study and how to tune those parameters.
  11. Conclusions. The current version of the conclusion mainly repeats and summarizes the background and methodizes, missing important discussion on authors' findings, contributions, and potential future works. 

Author Response

Dear Reviewer,
Thank you for taking your time and contributing to the quality of the presentation of our research.
We have made corrections to the paper. We hope that you will find our paper (after the corrections) more clear and professional.
Best Regards,
Authors
